# Choice Architecture Cueing to Healthier Dietary Choices and Physical Activity at the Workplace: Implementation and Feasibility Evaluation

**DOI:** 10.3390/nu13103592

**Published:** 2021-10-14

**Authors:** Eeva Rantala, Saara Vanhatalo, Tanja Tilles-Tirkkonen, Markus Kanerva, Pelle Guldborg Hansen, Marjukka Kolehmainen, Reija Männikkö, Jaana Lindström, Jussi Pihlajamäki, Kaisa Poutanen, Leila Karhunen, Pilvikki Absetz

**Affiliations:** 1VTT Technical Research Centre of Finland, Tietotie 2, P.O. Box 1000, 02044 Espoo, Finland; saara.vanhatalo@vtt.fi (S.V.); marjukka.kolehmainen@uef.fi (M.K.); kaisa.poutanen@vtt.fi (K.P.); 2Institute of Public Health and Clinical Nutrition, University of Eastern Finland, P.O. Box 1627, 70211 Kuopio, Finland; tanja.tilles-tirkkonen@uef.fi (T.T.-T.); markus.kanerva@laurea.fi (M.K.); reija.mannikko@terveystalo.com (R.M.); jussi.pihlajamaki@uef.fi (J.P.); leila.karhunen@uef.fi (L.K.); pilvikki.absetz@tuni.fi (P.A.); 3Finnish Institute for Health and Welfare, P.O. Box 30, 00271 Helsinki, Finland; jaana.lindstrom@thl.fi; 4D Department, Tikkurila Campus, Laurea University of Applied Sciences, Ratatie 22, 01300 Vantaa, Finland; 5Department of Communication, Business & Information Technologies, Universitetsvej 1, Roskilde University, 4000 Roskilde, Denmark; pgh@ruc.dk; 6Department of Medicine, Endocrinology and Clinical Nutrition, Kuopio University Hospital, P.O. Box 100, 70029 Kuopio, Finland; 7Faculty of Social Sciences, Tampere University, Arvo Ylpön katu 34, 33520 Tampere, Finland

**Keywords:** workplace, health promotion, prevention, type 2 diabetes, implementation research, behaviour change, choice architecture, nudge, diet, physical activity

## Abstract

Redesigning choice environments appears a promising approach to encourage healthier eating and physical activity, but little evidence exists of the feasibility of this approach in real-world settings. The aim of this paper is to portray the implementation and feasibility assessment of a 12-month mixed-methods intervention study, StopDia at Work, targeting the environment of 53 diverse worksites. The intervention was conducted within a type 2 diabetes prevention study, StopDia. We assessed feasibility through the fidelity, facilitators and barriers, and maintenance of implementation, building on implementer interviews (*n* = 61 informants) and observations of the worksites at six (t1) and twelve months (t2). We analysed quantitative data with Kruskall–Wallis and Mann–Whitney U tests and qualitative data with content analysis. Intervention sites altogether implemented 23 various choice architectural strategies (median 3, range 0–14 strategies/site), employing 21 behaviour change mechanisms. Quantitative analysis found implementation was successful in 66%, imperfect in 25%, and failed in 9% of evaluated cases. These ratings were independent of the ease of implementation of applied strategies and reminders that implementers received. Researchers’ assistance in intervention launch (*p* = 0.02) and direct contact to intervention sites (*p* < 0.001) predicted higher fidelity at t1, but not at t2. Qualitative content analysis identified facilitators and barriers related to the organisation, intervention, worksite environment, implementer, and user. Contributors of successful implementation included apt implementers, sufficient implementer training, careful planning, integration into worksite values and activities, and management support. After the study, 49% of the worksites intended to maintain the implementation in some form. Overall, the choice architecture approach seems suitable for workplace health promotion, but a range of practicalities warrant consideration while designing real-world implementation.

## 1. Introduction

Considering our susceptibility to external influences, changing behaviours requires targeting the contexts and environments in which behavioural decisions take place [1]. Workplaces provide an excellent setting for such interventions, as most adults spend a considerable share of waking hours at work. Workplace health promotion holds promise to benefit both employees and employers, for example, through improved employee wellbeing and productivity, reduced absenteeism and occupational health care costs, as well as enhanced corporate image and performance [2,3,4]. Societies, in turn, benefit through higher tax revenue and reduced social security costs because healthy workforces typically have better employment prospects, longer careers, and a higher income [5].

Health promotion has largely appealed to people’s conscious reflection by using educational approaches to guide individuals towards healthier behaviours [6,7]. The impact of such interventions has proven modest, however [8,9]. Suggested explanations include the automatic nature of much of human behaviour [10,11], and the imperfect rate at which beliefs and intentions convert into action [8,12]—particularly if the environment fails to support these intentions. Educational approaches also tend to favour socioeconomically advantaged individuals; hence bearing a risk of increasing health inequalities [13,14,15].

Environmental interventions that cue healthy behaviours primarily via automatic mental processes could yield effects with less cognitive effort, and independent of individuals’ socio-economic background and self-regulatory capacities [16,17]. Such interventions are closely tied with the concepts of nudge and choice architecture. Nudges encourage better choices by exploiting the known boundaries, biases, and routines of cognitive processes [18], the very features often preventing people from behaving rationally in ways that promote their own interests. In practice, nudges attempt to influence behaviour by modifying the surrounding choice architecture—i.e., the way that available choice options are presented in decision-making contexts—in ways that work independently of limiting the freedom of choice, substantially changing incentives, or relying on education [18,19]. Nudges typically work by reducing effort and cognitive load, increasing salience and attractiveness, or leveraging social norms [20]. Over a decade of intensive research [21], choice architecture interventions have proved effective in guiding food choices, for example, by altering food availability, position, order, and portion size [22,23,24,25], as well as by prompting healthier choices at the point of choice [26,27]. Physical activity, in turn, has increased through enhanced movement opportunities and contextual prompts [28,29].

Implementing choice architecture interventions is considered less resource-intensive compared to individual-level interventions [20,30]. Hence, scaling up to population level could be feasible [31]. Some evidence speaks for the feasibility of implementing prompting and proximity strategies in grocery shops to encourage healthy purchases [32], and digital decision-support systems in pharmacies to increase vaccination rates [33]. By contrast, in food service settings, scaling up a default type “dish of the day” strategy for promoting plant-based meals appeared challenging and yielded mixed results that depended on the context and target population [34,35,36,37]. However, overall evidence remains scarce on the implementation and feasibility of choice architecture interventions in real-world settings [20,38], including workplaces [39,40].

Impactful interventions are of little use, unless we know how to implement them effectively [41]. Studying implementation is thus necessary. Important elements of implementation process evaluation include the fidelity, barriers and facilitators, and maintenance of implementation [42,43]. Fidelity reflects the extent to which implementation follows plans [44], and reveals the likelihood with which interventions can and will be implemented successfully [45]. Besides projecting feasibility, assessing fidelity also supports accurate interpretation of study outcomes [39,46], as it enables determining whether the found effects—or lack of them—are due to the intended intervention or variations in its implementation [47,48]. Knowledge on fidelity also strengthens understanding of why interventions succeed or fail; thus, informing intervention development and optimisation [49,50]. The same rationale applies to studying contextual factors that may facilitate or hamper implementation and hence influence intervention effects [43,49]. Maintenance, in turn, refers to the extent to which implementation sustains over time [42] and serves as an important indicator of the overall feasibility and success of implementation.

In summary, restructuring the choice architecture appears an effective and equitable approach to support the adoption of healthy behaviours. However, research has nearly exclusively focused on impact assessment, leaving unanswered questions on implementation and feasibility. The current paper portrays the real-world implementation and feasibility evaluation of a choice architectural intervention designed to promote healthier dietary choices and physical activity at the workplace. The feasibility evaluation focuses on the fidelity, facilitators and barriers, and maintenance of implementation. In addition, items that are considered include the applicability to diverse worksites, ease of implementation, and required purchases of applied choice architectural strategies.

## 2. Materials and Methods

### 2.1. Study Design

We conducted a 12-month quasi-experimental pretest–posttest intervention, StopDia at Work, in natural settings at workplaces in three regions of Finland. The intervention took place between 2017 and 2019 within a larger type 2 diabetes prevention study, Stop Diabetes (StopDia) (Trial registration: NCT03156478) [51]. This study had the approval of the research ethics committee of the hospital district of Northern Savo.

The aim of the StopDia at Work intervention was to promote healthy dietary choices and daily physical activity at the workplace, with subtle modifications to the worksite environment, including common working spaces, personal workstations, recreation rooms, stairwells, elevators, and cafeterias. The employees of intervention sites received general information on the StopDia study and the collaboration between their workplace and the study. However, the employees were not disclosed the specific aim of the StopDia at Work intervention, that it is to alter workplace choice architectures to promote healthy behaviours mainly via automatic cognitive processes. This non-disclosure was to ensure the intervention would not inadvertently enhance employee self-awareness, prompt monitoring of the worksite environment, and stimulate a deliberate reflection of behavioural choices; hence interfering with employees’ natural responses to the intervention. 

### 2.2. Recruitment of Participating Organisations 

Through web searches and by consulting local ELY centres (Centres for Economic Development, Transport, and the Environment), we identified major public and private sector organisations operating in three regions of Finland. The three regions—Northern Savo, Southern Karelia, and Päijät-Häme—were the target areas of the StopDia study. The focus was on organisations with at least 100 employees and physical working environments suitable for the intervention. We contacted the management and/or human resources (HR) of potentially eligible workplaces (*n* = 86) via email and/or telephone, and arranged workshops (*n* = 4) for those initially interested in the study (Figure 1). Representatives of 31 organisations attended the workshops. In the workshops, these representatives discussed measures that workplaces had taken to promote employee health, as well as the potential facilitators and barriers of workplace health promotion. The representatives also received information on the choice architecture approach and brainstormed how to apply this approach to the workplace. After the workshops, we had additional one-to-one discussions with 23 volunteer workshop participants to further discuss the themes covered in the workshops. Workshop participants (*n* = 27) that expressed interest in the study, and organisations that had shown initial interest but were unable to send representatives to the workshops (*n* = 14), received an invitation to participate in the StopDia at Work intervention and a leaflet of the *StopDia Toolkit for Creating Healthy Working Environments* (Section 2.3). The leaflet introduced the choice architecture approach and a selection of practical strategies that had potential for implementation in the intervention.

Sixteen organisations with altogether 53 worksites decided to participate in the intervention (Figure 1). Each organisation chose one or more members of their personnel as implementers. These implementers were in charge of maintaining the intervention after the launch. In addition, the sites could have organisation-level coordinators that acted as contact persons between the research team and the intervention sites. Regarding 30 (57%) sites, our primary contact persons worked at the intervention sites, and the research team members visited the sites at least once during the intervention process. In the remaining sites, we communicated with organisation-level coordinators without actually visiting the sites. The coordinators and implementers typically represented HR or middle- or operational-level management, yet involved employees and cafeteria personnel as well.

### 2.3. Intervention Development and Content

As the basis of the StopDia at Work intervention, we developed the StopDia Toolkit for creating healthy working environments (Appendix A). This hands-on instrument is based on a comprehensive literature review and describes 53 practical strategies targeting generic workplace choice architectures, such as cafeterias, coffee rooms, and stairs. The strategies aim to facilitate healthier choices for diet and physical activity. The strategies were designed to be adaptable to diverse worksite environments, capable of reaching numerous employees within the workplace, and relatively effortless and inexpensive to implement. The toolkit applies both scientific literature and empirical knowledge to foster the adoption of dietary [52,53] and physical activity [54] guidelines for promoting health and preventing the development of type 2 diabetes and other lifestyle-related non-communicable diseases. Informed by the dual process theories that specify distinct reflective and automatic cognitive processes [10], we based the intervention mainly on automatic processes and applied the choice architecture approach [18,19,55]. We defined the toolkit strategies using three frameworks for applying behavioural insights: TIPPME [56], MINDSPACE [57], and EAST [58]. At an empirical level, the toolkit considers the needs and challenges of workplace health promotion identified through the recruitment phase discussions with contacted organisations (Section 2.2). Appendix A details the development and theoretical background and presents the full version of the toolkit.

Section 3.2 presents the toolkit strategies selected for implementation in the intervention, and details the applied behaviour change mechanisms, ease of implementation, and required purchases. We defined ease of implementation as the amount of knowledge and effort required to maintain a strategy after its launch. Easy strategies require little specialised knowledge, and besides occasional check-ups, no maintenance after launch. Moderate strategies require some knowledge on correct implementation and light maintenance on a regular basis, whereas demanding strategies require more specialised knowledge and daily maintenance. Required purchases suggestively indicate the extent to which implementation requires the procurement of new materials or services. Strategies with no purchases require no procuring, or in the case of this intervention, the study provided and delivered needed materials. Minor and substantial purchases refer to relatively inexpensive and relatively expensive goods, respectively.

### 2.4. Implementation Process

Preparations for implementation proceeded in collaboration with the coordinators and/or implementers and with the consent of the management of participating worksites. Thus, we consider that the researchers, implementers, and coordinators together acted as the choice architects of the study.

We became acquainted with the worksites through discussions with the coordinators and/or implementers, and visited sites accessible to us (*n* = 30) to map out opportunities for choice architectural modifications. Based on these discussions and visits, the research team and the coordinators and/or implementers selected intervention strategies from the toolkit (Section 2.3) individually for each site, and tailored the implementation of selected strategies to local contexts. Such contextualisation was justified, since the worksites (Section 3.1) were highly heterogeneous in terms of facilities, resources, and employees’ needs concerning diet and physical activity. The contextualisation involved planning of schedules, people involved, actions and materials needed, as well as physical spots to be adapted in the worksite environment. To maintain fidelity, we carefully recorded all adaptations and ensured the adaptations maintained the essential elements of the intervention [41,49]. These elements included, for example, using the same materials and placement principles, although targeted worksite environments and the form and delivery channels (print vs. electronic) of intervention materials varied across sites. Participation was free of charge for the organisations, and the study provided intervention sites with print intervention materials, such as posters and signs. However, should the sites choose to implement strategies that require the procurement of other materials, such as water bottles, height-adjustable desks, or gymnastic balls, the sites were responsible for the acquisition.

Intervention sites received illustrated instructions on the implementation of selected strategies. In 21 (40%) sites, researchers assisted the implementers and/or coordinators to launch the intervention. In the remaining 32 (60%) sites, the sites launched the intervention independently. The coordinators and implementers were asked to inform employees about the collaboration with the StopDia study and about provided intervention materials, as well as to encourage employees to use these materials. The employees were not, however, disclosed the specific aim of the intervention to alter workplace choice architectures to promote healthy behaviours predominantly via automatic cognitive processes.

After intervention launch, the sites independently maintained the implemented strategies over 12 months. Regarding one strategy that required weekly maintenance (#15, Section 3.2) and that all intervention sites intended to implement, implementers received checklists that they should sign each time they completed the maintenance. This procedure aimed to enhance implementation fidelity and to support fidelity assessment. In the Northern Savo region, implementers also received weekly text message reminders for this strategy, if they wished so. Implementers of 12 (33%) sites in this region opted for the reminders.

Where feasible, researchers or coordinators made follow-up visits to the intervention sites at month six (*n* = 41 sites; 77%) and month twelve (*n* = 18 sites; 34%). When visiting the sites was not possible, researchers conducted the follow-ups by phone. Besides supporting data collection and fidelity assessment, the follow-up sessions provided opportunities to enhance implementation. We answered implementers’ questions, encouraged implementers to maintain the intervention, and if needed and possible, helped to enhance the displayed intervention materials.

### 2.5. Data Collection

We collected data with several methods. Our primary data collection means were semi-structured interviews and observation. As complementary data, we collected photos from intervention sites, checklists returned by implementers (*n* = 21), and email and text messages exchanged with the coordinators and implementers. Post intervention, we requested additional information from sites with incomplete data via email and/or phone. In this paper, we refer to individual organisations with the capital letter O and identification numbers 1–16 (e.g., O1). Small letters following the organisation identifier indicate the worksites within the organisations (e.g., O1a).

#### 2.5.1. Interviews

The first two authors (E.R., S.V.) conducted the interviews over the follow-up visits and/or phone calls at months 6 and 12 (Section 2.4). These authors had a major role in the recruitment, intervention development, and implementation phases (Section 2.2, Section 2.3, Section 2.4), and they had thus become acquainted with the worksites as well as established rapport with the contact persons. The median durations of the first and second follow-up sessions were 60 min (range 20–180) and 30 min (range 20–120), respectively.

One organisation (O5) completed the intervention after six months, because its sites moved to new premises (Figure 1). Regarding this organisation, the first interview serves as the primary data on implementation. In another organisation (O11), two sites (O11b–c) completed the intervention after nine months because the sites, being construction yards, were closed (Figure 1). At these sites, the second interview took place shortly before the closing of the sites. At one site (O10a), the implementer was not available at month 6, and so the two interviews were merged and conducted at month 12. Sites O12c–q were not accessible to externals, and hence the organisation-level coordinator visited these sites after six months to check their implementation status.

At the follow-up visits, informants were interviewed in person, often at their personal workstations, and sometimes while they were performing their work tasks. In open and shared workspaces, personnel not involved in the implementation could be present as well. When visiting the intervention sites was not feasible, we conducted the interviews on the phone. The interviews ranged from individual to group interviews, depending on the number and availability of persons involved in the implementation. The researchers made notes during the interviews and typed the notes up as soon as possible after the interviews, while the discussions were still fresh in their minds.

The interviews involved altogether 61 informants, the majority of whom were females (*n* = 44). The informants represented predominantly implementers (*n* = 40) and coordinators (*n* = 11). However, some information was received from other informants (*n* = 10) as well. The informants represented professionals from numerous fields and both employees (*n* = 34) and managers (*n* = 19) of the participating organisations. Among the informants were, for example, HR personnel, occupational health and safety representatives, shop stewards, site managers, assistants, and cafeteria personnel. Two informants were external stakeholders of one participating organisation (O3), and the job titles of six informants remained unknown. Most informants (64%) had become acquainted with the interviewers over the planning and/or launch of the intervention, and were aware of the main purpose of the intervention.

The first interview covered questions on strategies that had been implemented, perceived success in launching and maintaining implemented strategies, if and how employees had been informed of and encouraged to tap into implemented strategies, possible difficulties encountered and ways of solving these difficulties, as well as perceived facilitators for and barriers to maintaining the intervention. In addition, we enquired about factors that motivate and do not motivate the implementers to maintain the intervention, and about persons most suitable for the implementer’s role. The second interview asked about changes in the implementation since the first interview and enquired about whether the sites intended to maintain the intervention after the study.

#### 2.5.2. Observation

When feasible, we made quality assurance tours in the worksite environments during the follow-up visits. The purpose of these tours was to record observations on the quality of implementation and thus to complement data collected with interviews. The tours covered altogether 39 (74%) worksites (t1: *n* = 37 (70%), t2: *n* = 13 (25%)), representing the majority of intervention sites. Such a well-selected sample is considered capable of providing sufficient insight on implementation [43]. The first two authors (E.R., S.V.) conducted the tours and recorded observations as field notes and/or photos. The only exceptions were sites O12c–q that were not accessible to externals and that were toured by the organisation-level coordinator.

### 2.6. Analyses

We used NVivo R1 (QRS International) to manage and analyse qualitative data, and Microsoft Excel^®^ 2016 (Redmond, WA, USA) and IBM SPSS^®^ Statistics 25 (Armonk, NY, USA) for quantitative data.

#### 2.6.1. Fidelity

We assessed fidelity both qualitatively and quantitatively, focusing on the dose delivered and the quality of implementation. We measured dose as the number of practical strategies implemented per site and evaluated implementation quality against an assessment framework (Appendix A) and site-specific implementation plans. The quality assessment framework was developed in this study and comprises the essential elements of and a tripartite assessment scale (2 = successful, 1 = imperfect, and 0 = failed) for each implemented practical strategy. Evaluating the quality of implementation categorically has also been common in prior implementation research [41].

Qualitative analysis: We compiled all available data on implementation at intervention sites and organised the data according to the site, strategy, and follow-up time point (t1 = month 6, t2 = month 12). We performed the implementation quality assessment individually for each strategy at each site and at each time point, and refer to this unit of analysis as “case”. Two authors (E.R., S.V.) independently rated the quality of all cases, discussed and agreed on differing ratings, and consulted a third author (P.A.) in uncertain cases. The assessment process comprised several rating and discussion rounds, along which we refined the assessment framework and the definitions of implemented strategies as well as requested further details from sites with incomplete data. Across all assessment rounds, the mean interrater agreement was 89%. Cases with too little data available for reliable quality assessment received a code N/A. The assessment focused on toolkit strategies launched during the intervention and excluded strategies that the participating worksites had adopted already before the intervention.

Quantitative analysis: Pooling all intervention sites, implemented strategies, and follow-up measurements, our dataset comprised 412 individual cases (t1 = 209, t2 = 203). Within this sample, 75 cases (t1 = 22, t2 = 53) were coded N/A due to incomplete data. Thus, 337 cases (t1 = 187, t2 = 150) received implementation quality ratings and were included in statistical analyses. The rated cases covered 82% (t1 = 90%, t2 = 74%) of the full sample. Of the cases coded N/A, 95% (t1 = 82%, t2 = 100%) concerned sites to which we had no direct contact and 100% represented strategies that the sites implemented independently without researchers’ assistance. In addition, all N/A cases represented sites that received no text message reminders (Section 2.4) for strategy #15 (Section 3.2).

Using the cases that received implementation quality ratings, we examined whether these ratings were dependent on four independent variables: (1) the ease of implementation of applied strategies, (2) researchers’ assistance in intervention launch, (3) direct contact to intervention sites, and (4) sending text message reminders to implementers. We assessed these associations separately for ratings at six (t1) and twelve months (t2). Statistical tests of normality, Shapiro–Wilk and Kolmogorov–Smirnov, indicated that across the four independent variables and at both time points, the implementation quality ratings did not follow a normal distribution (*p* < 0.05). Hence, we employed nonparametric statistical tests [59], defining *p*-values < 0.05 as statistically significant and reporting all *p*-values as two-tailed. Independent samples Kruskall–Wallis test assessed the difference in implementation quality ratings between the three levels of implementation ease: easy, moderate, and demanding. Independent samples Mann–Whitney U test assessed the difference in implementation quality ratings between cases that received and cases that did not receive researcher’s assistance, direct contact, or reminders.

#### 2.6.2. Facilitators and Barriers of Implementation

We explored the facilitators and barriers of implementation with descriptive qualitative content analysis [60]. We performed the analysis from a factual perspective, assuming that our informants had answered the interview questions to the best of their knowledge, and that the data they had shared reflected reality more or less truthfully [61]. Due to the practical orientation of this work, the analysis focused on the visible and obvious content of collected data (i.e., *manifest content*), instead of interpreting underlying meanings hidden between the lines (i.e., *latent content*) [62]. We adopted a deductive approach in that we employed a framework proposed for grouping facilitators and barriers of workplace health promotion interventions [39,42]. This framework comprises five domains that distinguish between the characteristics of (1) the socio-political context, (2) the organisation, (3) the implementer, (4) the intervention, and (5) the participant, referring to the subjects of the intervention [39,42]. Since our analysis identified no facilitators nor barriers related to the socio-political context, we excluded this domain from the framework. Instead, we identified facilitators and barriers related to the worksite environment and included an additional domain: “physical and digital environment”. To avoid confusion with participating worksites, we labelled the domain “participant” as “user”. Hence, our final categorisation matrix involved the following domains: (1) organisation, (2) intervention, (3) physical and digital environment, (4) implementer, and (5) user; user referring to the employees of intervention sites who became exposed to the intervention. We systematically coded the data according to these domains, and within each domain, generated categories freely following the principles of inductive qualitative content analysis [60].

The first author (E.R.) immersed herself in the data through reading and rereading, simultaneously coding the data and organising similar codes under higher-order headings or categories. The validity and reliability of the coding was ensured through a peer-checking process, common in qualitative research [63,64]. This meant that the first author iteratively reviewed a sample of codes and their corresponding raw text with three other authors (S.V., P.A., and L.K.), and the four authors refined and agreed on the codes and their grouping into categories and domains.

#### 2.6.3. Maintenance

We measured maintenance as the proportion of intervention sites that intended to maintain at least one implemented strategy after the study. By participating in the study, the intervention sites agreed to sustain implemented strategies over 12 months. Continuing implementation longer than this was thus not expected. In the 12-month follow-up interview (Section 2.5.1), we nevertheless enquired whether the sites intended to continue implementation. In addition, while requesting additional information from sites with incomplete data on intervention delivery over the 12-month study, we received some information on post-study maintenance as well.

## 3. Results

### 3.1. Participating Organisations

Table 1 presents the characteristics of the 16 participating organisations. Three of the organisations operated in the region of Southern Karelia, four in Päijät-Häme, and nine in Northern Savo. The organisations represented both private (*n* = 10) and public sector (*n* = 6), and various fields of operation. From each organisation, 1−20 (mean 3.3) distinct worksites or departments were involved in the intervention, forming the study sample of altogether 53 intervention sites. Among these worksites were grocery shops, factories, a university of applied sciences, bureaus, a farm, a kindergarten, construction yards, hospital departments, and a welfare services centre. Nine organisations had worksite cafeterias on intervention sites, and four of these organisations involved the cafeterias in the intervention. Over 5000 employees in total worked at the intervention sites (Figure 1), and the proportion of male employees within organisations ranged from 5 to 91% (mean 43%). In 12 organisations, the work ranged from sedentary to physical, whereas in four organisations the work was predominantly sedentary. In ten organisations, at least part of the employees worked in shifts.

### 3.2. Characteristics of Implemented Strategies

#### 3.2.1. Descriptions, Mechanisms, and Settings

Table 2 portrays the characteristics of the practical strategies implemented in the intervention. In total, 23 strategies were launched by at least one intervention site, representing 43% of the strategies included in the toolkit (Appendix A). Of these strategies, 16 promoted nutrition and seven physical activity. Overall, the implemented strategies applied 21 diverse behavioural change mechanisms. Implementation settings comprised coffee rooms, cafeterias, meetings, personal workstations, common environments, stairs, and elevators.

The three most often implemented strategies were *#15 encourage smart packed lunches*, *#20 prompt context-specific movement*, and *#16 encourage provision of fruit at work* (Table 2). These strategies were implemented at 48 (91%), 43 (81%), and 17 (32%) sites, respectively. At 31 (58%) sites, the entire intervention consisted of one or more of these three strategies. Strategy #15 comprised a year-long packed lunch of the week recipe campaign that primed the preparation of nutritionally high-quality packed lunches. This strategy aimed to cultivate descriptive social norms of what packed lunches could be, and to break up the complex behaviour of healthy eating into more manageable and attractive tasks. Strategy #20 aimed to prompt context-specific movement with a series of Flex! movement posters depicting simple movements suitable to be performed within daily work tasks. Strategy #16 provided a starting kit for forming fruit crews, i.e., social circles in which the members take turns to organise fruit provision at work. This strategy aimed to tap into social networks and people’s inclination for reciprocity, to encourage commitment contracts, and to cultivate the social norm of offering healthier food at the workplace. In two grocery shops (O14a–b), the implementation of this strategy was adapted so that the employer provided the fruit and the workers of the fruit and vegetable section arranged regular fruit offerings in staff coffee rooms. For images of the materials of these three strategies, see Appendix A.

Fifteen (65%) strategies were each launched by less than five intervention sites (Table 2). Of these strategies, 12 were related to nutrition and required some sort of food offering at the intervention site. These twelve strategies were implemented at sites (*n* = 6 in total) that had on-site cafeterias involved in the intervention and/or that often organised meetings with food and beverage provision.

Intervention sites mainly kept to their implementation plans and enacted strategies that were selected in the designing phase (Section 2.4). Eight sites, however, ended up implementing one or two additional strategies from the toolkit alongside their originally planned strategies. These so-called spin-off strategies are included in Table 2, and concerned strategies #1, 10, 17, 21, 22, and 23.

#### 3.2.2. Ease of Implementation

According to our definition (Section 2.3), ten (43%) of the implemented strategies were categorised as easy to maintain, nine (39%) moderate, and four (17%) demanding (Table 2). The three most often implemented strategies (#15, 20, and 16) were easy to moderate to maintain. Easy strategies mainly increased the availability of opportunities that enable healthy behaviours, and used contextual cues that encourage such behaviours. The availability increased through providing employees reusable water bottles, fruit, light exercise equipment, a break exercise application, and/or wobble chairs. The contextual cues, in turn, prompted movement and/or stair use or primed healthy food choices.

The moderate to demanding strategies focused largely on nutrition and were delivered in cafeterias and meetings. These strategies altered the availability, salience, accessibility, convenience, and/or size of food options, as well as prompted choosing healthier options. Maintaining these strategies typically required knowledge on the nutritional quality of foods and constant maintenance because the food choice architecture keeps changing as people choose and consume foods.

#### 3.2.3. Required Purchases

Sixteen (70%) of the applied strategies required no purchases, six (26%) required minor, and one (4%) substantial purchases (Table 2). For the three most often implemented strategies (#15, 20, and 16), the study provided required materials. Minor purchases comprised food products procured to cafeterias or meetings, as well as reusable water bottles, light exercise equipment, and a break exercise application provided for employees. Substantial purchases involved wobble chairs acquired for common work environments.

### 3.3. Fidelity

#### 3.3.1. Dose Delivered

Except for one site, all intervention sites implemented at least one strategy. The median number of implemented strategies was three (range 0–14), with a median of two strategies (range 0–9) promoting nutrition, and one strategy (range 0–5) promoting physical activity. The number of implemented strategies differed, however, between sites that had on-site cafeterias involved in the intervention and sites that had no participating cafeterias. At sites with cafeterias (*n* = 4), the median number of implemented strategies was 10.5 (range 9–14), with a median of 8.5 (range 8–9) strategies related to nutrition and two (range 1–5) strategies related to physical activity. In contrast, sites with no cafeterias (*n* = 43) implemented a median of three (range 0–7) strategies; two (range 0–4) focusing on nutrition and one (range 0–4) on physical activity.

#### 3.3.2. Quality of Implementation

Implementation quality was rated for 187 cases at month 6 (t1) and for 150 cases at month 12 (t2). *A case refers to a given strategy implemented at a given worksite at a given follow-up time point*. Figure 2 presents the distribution of implementation quality ratings by implemented strategy and follow-up time point. Overall, implementation was successful in an average of 66% (t1: 64%; t2: 69%), imperfect in 25% (t1: 26%, t2: 23%), and failed in 9% (t1: 11%, t2: 7%) of the rated cases.

We examined the association of implementation quality with the ease of implementation (Section 2.3), researchers’ assistance in intervention launch, mode of contact to the intervention sites, and text message reminders (Section 2.4) received (Table 3). Ease of implementation was not statistically significantly associated with the quality of implementation at either time point (t1: *p* = 0.54, t2: *p* = 0.19). Researchers’ assistance (*p* = 0.02) and direct contact to intervention sites (*p* < 0.001) were associated with higher implementation quality at t1, but the associations disappeared at t2 (*p* = 0.63 and *p* = 0.98, respectively). Receiving reminders had no statistically significant association with implementation quality at either time point (t1: *p* = 0.10, t2: *p* = 0.29).

The majority of rated cases were categorised as easy to implement (t1: 53%, t2: 45%), followed by cases that were moderate (t1: 40%, t2: 46%), and demanding (t1: 7%, t2: 9%). In slightly over one third of the rated cases (t1: 34%, t2: 36%), researchers had assisted intervention launch, and in nearly three thirds of the cases (t1: 68%, t2: 78%), communication to the intervention sites had been direct. Weekly text message reminders for strategy #15 (Table 2) were received at slightly over one third of the rated cases (t1: 32%, t2: 38%).

### 3.4. Facilitators and Barriers of Implementation

Across the five domains of the used categorisation matrix (Section 2.6.2), our qualitative content analysis identified 11 main categories of facilitators and 12 main categories of barriers (Figure 3). Both facilitators and barriers included categories related to the characteristics of the organisation, intervention, physical and digital environment, and implementer. Barriers also comprised one category related to the user.

#### 3.4.1. Facilitators

##### Characteristics of the Organisation

Careful planning was a major organisational facilitator that involved clear division of responsibilities, communication, sufficient resourcing, and integration into existing health promotion activities. Several informants highlighted the importance of dividing responsibilities clearly (O3, O10b, O13, and O14a–b), and the implementation rolled out smoothly at sites that explicitly defined who should do what (O10b, O13, O14a–b). As for communication, informants (O3, O11e) considered it recommendable to formulate a communication plan and inform employees of the intervention. One implementer (O11e) thought it would be helpful if employees were aware of implemented strategies and the reason for, for example, changed food provision in meetings, “*So they wouldn’t think the changes were from me*”. Informants also stressed the necessity of ensuring sufficient resources, including enough time for planning the launch and maintenance of intervention strategies (O3, O2). The significance of integrating the intervention into existing operations was crystallised by one coordinator (O1): “*Chances for success are higher when linked with the organisational context. If the initiative is not connected to the main activities, it easily remains undone*”. Another coordinator (O13) provided a successful example of how this integration materialised: “*The launch of the intervention occurred at a good time, because an ongoing wellbeing initiative had discussed, inter alia, nutrition and sleep, so the strategies served as good support measures for the ongoing initiative*”.

Another organisational facilitator was management engagement. The management can support implementers by participating in the implementation and encouraging employees to tap into provided opportunities (O11e).

##### Characteristics of the Intervention

The intervention afforded utility to the implementer, manifested in opportunities for breaks and physical activity (O7, O8, and O11c), as well as in food for thought (O5c, O10a, and O12b). As for breaks, one implementer (O11c) said: “*Changing the recipes (#15) breaks the workday and you get to stretch the legs*”. Regarding food for thought, one implementer (O12b) portrayed how understanding of the rationale behind the intervention sparked motivation for implementation: “*The study woke me to think of type 2 diabetes and that I wouldn’t want to get it. That raised my interest in nudging as well*”.

Compatibility with the worksite denotes that the intervention fits the mission of the worksite and the work of the implementer. This theme relates to the organisational facilitator “careful planning”, whereby the organisation can adjust and integrate the intervention into the organisational context. Reflecting fit with the worksite mission, the head of one cafeteria (O12a) said: “*Serving health promoting food is the responsibility and the value of the cafeteria*”. Indicating fit with implementers’ work, informants from several sites (O1, O11b, O12a, O13, O15, and O16) reported that the implementation could be integrated into the duties of the implementer. One coordinator (O13) portrayed how the maintenance of the recipe campaign (#15, Table 2) fits the work of their occupational health and safety (OHS) representative: “*The duties of the representative include a weekly tour in the working environments, and changing the recipe cards could be integrated into this tour*”. At another site (O15), the same strategy supported the OHS representative to perform the representative’s role: “*Visits to coffee rooms enable meeting the personnel in person, discussing the recipes or other matters, and meeting new employees. The recipes provide a reason to visit the workstations*”.

Reflecting the ease of maintenance, a number of informants described the intervention as easy, simple, natural, and/or effortless to maintain (O1b, O5a, O5c, O6, O10b–c, O11c, O11e, O12a–b, and O16). One implementer (O12b) also discovered that when displayed successfully, intervention materials per se remind them of their maintenance. Indicating perceived reach and effects, implementers found it motivating to observe how the intervention reaches employees (O5) and starts to take effect (O7). Finally, implementers were satisfied with the support received from the research team. This support involved co-design of implementation (O5b), fluent delivery (O5b, O13) and clear packaging of provided materials (O11c, O12b), as well as reminders sent for strategy #15 (Table 2) (O11c, O11e, and O14c).

##### Characteristics of the Physical and Digital Environment

Practical channels for distributing intervention materials within the worksite included internal mail (O2, O10b), info screens, email, and intranet (O11c, O13, and O15). Compared to delivering print materials, digital delivery was considered more effortless for the implementer, yet potentially inferior in reaching employees. One implementer (O15) justified this viewpoint as follows: “*Digital delivery would facilitate the dissemination of the recipes, but uploading the recipes on the intranet, for example, would require employees to go and get the recipes from there. In that case, it’s likely fewer would find them*”. Existing worksite food supply facilitated the implementation of many eating-related strategies. For example, sites with cafeterias and/or a custom to provide refreshments in meetings (O3, O7, O8, O11e, O12a, and O15) successfully applied a variety of strategies. In grocery shops (O14a–b), in turn, fruit stocks enabled arranging regular fruit provision in coffee rooms.

##### Characteristics of the Implementer

Characteristics of work that facilitated implementation comprised duties that involve regular touring of worksite premises (O2, O7, and O13), location at the intervention site (O1, O4, O6, and O10b–c), regular working hours (O12a), and time available for the implementation (O10c). Besides these practical aspects, many informants considered it also natural if the substance of implementers’ work relates to the intervention (O1, O3, O5, O6, O8, O9, O11e, O12a–b, O13, O14a–b, and O15). In our study, these criteria applied to HR personnel (O9), occupational health and safety representatives (O13), communication specialists (O3), cafeteria personnel (O12a), and workers of the fruit and vegetable section of grocery shops (O14a). According to informants, the implementer’s role suits both managers (O1, O8, O11a, O11d, O11e, O12a, and O14a) and employees (O6, O10a, and O12a).

Individual characteristics attributed to persons suitable for maintaining the intervention involved committed, motivated and motivational, relatable to employees, sociable, organised, and tolerant to employees’ initial resistance to change. Commitment manifested itself in the way that implementers conscientiously maintained the intervention regardless of their personal attitudes towards this task. For example, one implementer (O11c) said: “*The firm pays for working, and maintaining the intervention is part of the duties. I wouldn’t change the recipes for fun during free time*”. A coordinator (O14b) expressed similar thoughts: “*When you have involved yourself in the project and committed to the maintenance, you will do it*”. This coordinator pondered, however, that it would be beneficial to find an implementer who is motivated and motivational as well: “*The work community needs ambassadors that show the way with their own behaviour and inspire and encourage other employees to try out new things and change their behaviour*”. Furthermore, other informants mentioned the importance of motivation and interest in the intervention (O4, O7, O10c, O11a, O12b, O14a, O15, and O16). Related to being motivational, informants considered it beneficial if the implementer is close, or relatable, to employees (O4), and sociable (O11c).

Being organised appeared in the way that implementers created and used reminders for strategies that require active maintenance (O5, O9, O10b–c, O11b–c, and O16), integrated the maintenance into existing routines at the workplace (O9, O14c, O16), and performed maintenance tasks regularly. Consequently, several informants reported that the implementation became a routine (O10b, O11a, O11e, O13) that needs no reminding (O11a, O12b). Implementers demonstrated organisation also by enhancing the display of intervention materials (O10a, O13, and O15) and by arranging stand-ins (O9, O12o) if need be.

The ability to maintain the intervention despite negative feedback from employees was the key to success in one cafeteria (O7), where employees’ initial response to new arrangements (strategies 4–5 and 11; Table 2) was undesirable. Over time, however, the employees understood the purpose of the strategies to facilitate healthier food choices and portion sizes, and agreed with the changes. This occurrence links to the organisational facilitator “careful planning” and the finding that communicating the intervention to employees could facilitate implementation.

#### 3.4.2. Barriers

##### Characteristics of the Organisation

Lack of management support was a rare problem, concerning only one site (O11e). At this site, however, the issue bothered the implementer throughout the study, making them feel left alone with the implementation. Lack of resources manifested as a lack of time and personnel (O3, O4a–c, and O7). Typically, this issue was due to busyness with competing priorities, as one coordinator portrayed (O7): “*We are growing with a huge speed and are pretty much tied with the recruitment and orientation of new employees*”. Regarding organisational changes, one coordinator (O5) noted how “*all shifts and distractions in the routines of the organisation complicate implementation*”. A common example was staff turnover. When the implementer changed jobs, the implementation easily ceased (O5a–e, O11c–d) or the implementer’s role could pass on to the next implementer with deficient instructions (O10a). This issue of poor knowledge transfer links to the final organisational barrier, poor flow of information, which manifested at a few sites (O3, O4a–c, and O11e). Information flowed poorly from coordinators to implementers (O4a–c) or from coordinators and/or worksite management to employees (O3, O11e). Reasons for failed communication included scattered organisation structure (O4a–c) and the above-mentioned barrier: lack of management support (O11e).

##### Characteristics of the Intervention

Unclear implementer instruction is an issue that concerns both the intervention—and hence the researchers—and the organisation, and that relates to the organisational barrier of “poor flow of information”. Ensuring that everyone involved in the implementation receives sufficient information is crucial to fidelity, but it proved challenging, particularly in organisations with multiple intervention sites and/or implementers (O5), and in situations where the implementer changed (O10a). Suboptimal knowledge transfer bothered two implementers (O5b, O11e) that remained unsure of what was expected from them.

Intervention requirements that challenged implementation involved efforts, duration, and costs. Remembering to perform implementation tasks and to remind other implementers to perform theirs appeared challenging at first, but the burden of remembering reduced over time as the implementation “*fell into a routine*” (O10b). Maintaining the packed lunch recipe strategy (#15, Table 2) felt too burdening for one implementer (O1), and the 12-month duration too long for another (O2). Costs proved a barrier to sustained implementation at one site (O14b) that had chosen to implement the fruit crew strategy (#16, Table 2) by treating employees with unlimited fruit on every workday. In contrast, another site of the same organisation (O14a) found this strategy feasible by providing one fruit per employee twice a week. This example illustrates how intervention intensity can be adapted, and how adapting intensity allows adjusting costs.

The final intervention-related barrier, perceived ineffectiveness, terminated the maintenance at one site (O14a), where the implementer lacked motivation to maintain the recipe strategy (#15, Table 2) because “*the recipes did not seem to interest the employees*”.

##### Characteristics of the Physical and Digital Environment

Physical worksite environments limited implementation possibilities at a few sites. Finding feasible places and ways to display print intervention materials challenged implementation at two sites (O10a, O13). In cafeterias, fixed serving lines in which the arrangement of and space for various foods are unchangeable restricted the number of strategies that could be implemented and the way in which selected strategies could be delivered. The head of one cafeteria (O12a) reflected that “*a new serving line with separate salad bar and more room could promote healthy food choices*”, but at the time, such a substantial procurement was not on the agenda. Regarding digital environments, the delayed introduction of company’s internal social media platform prevented the digital delivery of intervention materials that had been planned at one site (O14a). Renovations, in turn, required the removal of all intervention materials and interrupted the implementation for several months at two sites (O10a, O15).

##### Characteristics of the Implementer

Characteristics of work that challenged implementation comprised irregular working hours, heavy workload, and a job substance unrelated to the intervention. Irregular working hours were problematic with strategies requiring regular maintenance (O10a, O12a), such as the packed lunch recipe campaign (#15, Table 2), because “*work days vary in shift work, and changing the recipes is not always possible on the same weekday*” (O12a). Irregular maintenance, in turn, complicates remembering and forming a habit of the implementation. Heavy workload manifested itself in the lack of time (O1, O10a, O11b, O11e, O12e, O14a, and O15) and in the declined coping of the implementer (O3). This issue links with the organisational barrier of “lack of resources”. A job substance not related to the intervention bothered two assistants (O5a-b), one of whom thought the implementation “*didn’t feel natural*” within their job (O5a).

Individual factors that hampered the implementation involved forgetting, absenteeism, and negligence of intervention materials, as well as the lack of motivation, personal relevance, and understanding of the intervention. As a minor problem, implementers reported occasional forgetting of maintenance tasks (O1b, O11a, and O12a). A major problem, in turn, was implementers’ long absences, which could cease the implementation over longer periods (O1, O14b). In such cases, arranging stand-ins was beneficial, as long as the stand-ins received sufficient instructions. Otherwise, the fidelity might decline, as happened at one site (O15a). The negligence of intervention materials manifested at two sites (O10a, O15), where the implementers failed to reintroduce materials removed due to renovations. The lack of motivation, personal relevance, and understanding of the intervention were barriers identified in one organisation (O4). The coordinator of this organisation portrayed how their implementers—the site managers—were “*very competitive and young, and might not find diabetes a personally relevant subject*”, and pondered that “*the managers might not see the connection between health promotion activities, diabetes, and, for example, absence from work*”. In this organisation, the implementers received minimal introduction to the intervention and little support for implementation, as the coordinator assigned the implementation responsibility via email. The above examples of insufficient stand-in introduction, negligence of intervention materials, and lack of understanding relate to the organisational barrier of poor flow of information and the intervention-related barrier unclear implementer instruction.

##### Characteristics of the User

The users of intervention materials challenged implementation, because they moved materials away from their assigned places. Materials disappeared (O7, O9, O10a), were thrown away over cleaning (O1PA), or were moved out of the way and hidden in cupboards (O10a). Exercise equipment travelled to employees’ personal workstations and under or behind furniture (O9, 15). On one hand, the moving of materials was a positive sign, indicating the materials were noted and used. On the other hand, mobility increased implementer burden, requiring implementers to collect and bring the materials back to where they belong.

### 3.5. Maintenance

As the final indicator of feasibility, we surveyed the maintenance of implemented strategies post study. Overall, we obtained maintenance information from 32 sites (60%). Of these sites, 26 (81%) kept maintaining, considered reintroducing, or planned to apply in a modified way at least one strategy. This continuation involved nutrition-related strategies implemented at cafeterias and meetings, the packed lunch recipe campaign (#15, Table 2), the fruit crew strategy (#16), and several strategies for physical activity (#17–22). Known reasons for discontinuation included the implementer leaving the site, the site being closed, the disposal of materials, and high implementation costs.

## 4. Discussion

Choice architecture—the variety, arrangement, properties, and presentation of choice options—can have a powerful, often unnoticeable influence on behaviour. The main emphasis of choice architecture research has been on effectiveness, while implementation and feasibility have remained less studied. We portrayed the implementation and feasibility evaluation of a 12-month choice architecture intervention at diverse worksites. The intervention employed a broad range of choice architectural strategies related to nutrition and physical activity. Implemented strategies were selected and contextualised individually for each site via bilateral dialogues between the research team and the worksites. Semi-structured interviews and observations indicated that implementation was successful at two thirds of evaluated cases, and prospects for maintaining implementation post study emerged at a substantial proportion of worksites. Implementation quality was independent of reminders and the ease of implementation of applied strategies, but researchers’ assistance in intervention launch and direct communication with implementers seemed beneficial within the first six months. Furthermore, an array of contextual factors influenced implementation.

### 4.1. Implementation and Feasibility Evaluation

#### 4.1.1. Applicability to Worksites, Ease of Implementation, and Required Purchases

All participating worksites found strategies suitable for their settings from the StopDia Toolkit for Creating Healthy Working Environments, the pool of strategies from which the ones implemented were selected. This indicates that the toolkit and choice architectural strategies in general serve diverse workplaces. The applicability of several nutrition-related strategies, however, was limited at worksites without cafeterias, vending machines, or other pre-existing food provision. At such sites, feasible nutrition strategies were restricted to the packed lunch of the week recipe campaign (#15) and the fruit crew-strategy (#16). These strategies encourage healthier food choices by increasing the salience and social acceptability of healthy foods, as well as by facilitating the availability of such foods. These strategies do not provide the encouraged foods there and then, however. For wider application of nutrition-related choice architectural strategies and to further reduce the amount of individual resources—or “agency” [7]—required for making healthy food choices during working hours, workplaces should make health-promoting foods available for their staff. Increasing availability would be justified, because the use of worksite catering services has proved to predict healthier dietary patterns among the working population [65,66,67]. Motivating workplaces to improve healthy food availability might require government policy actions, such as tax incentives or standards for food procurement [68,69,70,71,72]. In Denmark, for example, the government-launched Organic Action Plan 2020 has increased the procurement and hence availability of organic foods in public kitchens [73].

Choice architecture interventions are considered relatively effortless to implement [30,74]. Supporting this claim, we scored the majority of strategies implemented in this study and the majority of strategies in the StopDia Toolkit easy or moderate to implement, defined as requiring little specialised knowledge and light or no maintenance after launch. In line with this scoring, a number of implementers found the intervention effortless to maintain alongside work duties. Nevertheless, the choice architecture approach features also more challenging strategies, particularly within the nutrition domain. Yet, our results indicate that workplaces can successfully implement demanding strategies as well (#4–6 and 12), and that implementation quality is independent of how demanding a strategy is. Considering that our implementers represented diverse occupational groups without earlier experience in the choice architecture approach, learning the implementation seemed possible with the support that the research team provided. This support comprised the co-design of the intervention, illustrated instructions and on-site assistance for intervention launch, as well as follow-up visits to support sustained implementation.

Besides being effortless, choice architecture interventions are considered relatively inexpensive [6,20]. Our findings support this assumption in that the delivery of nearly all implemented strategies and the majority of strategies in the toolkit require no or minor purchases. Unsurprisingly, implementation sites also seemed to prefer these less expensive strategies, since only one site chose to implement a strategy that required a substantial purchase. Implementation costs are not restricted to purchases, however, but include implementer training too. Estimating the full costs of implementing choice architecture interventions, including training, fell out of the scope of the current paper, yet would be an important topic for future research.

#### 4.1.2. Fidelity

With a median of three implemented strategies per site and with two thirds of implementations evaluated as successful and one fourth partially successful, we consider the overall fidelity in this study satisfactory. According to a literature review on implementation studies, expecting perfect or near-perfect implementation is unrealistic and unnecessary because few interventions have reached implementation levels closer than 80% of optimal, and studies have yielded positive results with levels around 60% [41].

A few matters warrant consideration, however, while interpreting our fidelity findings. First, we were unable to rate the fidelity of 18% of all cases due to incomplete data, and decided to exclude these cases from statistical analyses. Importantly, the non-rated cases nearly exclusively represent sites that missed the three support measures that the research team could offer: direct communication, on-site assistance in intervention launch, and reminders. In addition, the number of excluded cases was substantially higher at twelve versus six months. These factors may have influenced the observations that direct contact and assistance predicted higher fidelity at six but not at twelve months, and that reminders had no significant association with fidelity. According to earlier research, technical assistance, such as efforts to support implementers to solve problems and maintain motivation and commitment is essential for effective implementation [41].

Two other remarks on our fidelity results concern the used assessment framework. First, since the framework comprises only three grades (successful, imperfect, and failed), it is rather insensitive to variations in implementation intensity, particularly at the higher end of the assessment scale. Hence, sites may have received equal grades with various levels of implementation intensity. For example, the packed lunch recipe campaign (#15) was rated as successfully delivered both at sites that distributed the materials through one channel (e.g., info screens), and at sites that used multiple channels (e.g., print materials in coffee rooms and digital distribution through info screens and email). In these examples, both delivery modes met our minimum criteria for successful implementation, although the multi-channel approach, which equals a higher dose, might prove more effective in reaching employees and influencing their behaviour [39]. Second, our fidelity ratings reflect both absolute implementation performance and performance relative to the site-specific implementation plans. This entails that equal performance sites with ambitious plans (e.g., several new products to worksite cafeterias) could receive poorer grades than sites with less ambitious plans (e.g., few new products to cafeterias).

#### 4.1.3. Facilitators and Barriers of Implementation

Our qualitative analysis indicated that successful implementation requires adjusting and integrating the intervention into the values, ongoing activities, and resources of the organisation; careful planning and resourcing; as well as a management that supports and actively engages in the implementation. These findings cohere with the results of prior workplace health promotion interventions [39,75], choice architecture studies in pharmacy [33] and retail settings [32], and intervention studies from other fields [41]. In addition, the results reflect the normalisation process theory (NPT) [76,77] and the diffusion of innovations theory (DIT) [78], which support understanding of how new practices become adopted and routinely embedded in social systems. According to both these theories, the compatibility of the intervention with the values, goals, and operations of the organisation is crucial for adoption [76,78]. This entails that while targeting generic choice architectures, such as workplace cafeterias or coffee rooms, and while employing strategies generally relevant for and applicable to these choice architectures, some level of contextualisation is often necessary for effective implementation. Fortunately, literature suggests that contextualisation and fidelity can coexist, given that interventions preserve their essential elements [41,49].

Related to our findings on careful planning, resourcing, and management support, NPT highlights the willingness and commitment of actors involved in the implementation to invest efforts in defining, organising, resourcing, and enacting needed procedures through *cognitive participation* and *collective action* [76,77]. We attempted to support such involvement by designing the intervention in collaboration with the participating worksites. Research suggests that shared decision making, which involves non-hierarchical relationships, mutual trust, and open communication between involved partners, is associated with superior and sustained implementation [39,41]. Shared decision making also reflects the interactive systems framework for dissemination and implementation (ISF), which emphasises the need for collaboration and two-way interaction between stakeholders involved in bridging research and practice [79].

Regarding the intervention, we found key facilitators to involve the perceived utility of the intervention to the implementer, as well as perceived ease of maintenance, reach, and effects. These facilitators align with DIT, which postulates that a rapid adoption requires perceiving the practice as relatively advantageous, easy to implement, and effective [78]. Similarly, literature reviews on implementation research have identified perceived benefits, ease, and effects to facilitate implementation [39,75]. Our results indicated, however, that strategies requiring regular maintenance might feel burdensome in the beginning—even with relatively effortless to implement strategies. This finding is unsurprising because remembering new tasks demands conscious effort [80,81,82]. Paradoxically, achieving choice architectures that guide healthy behaviours automatically requires the choice architects to learn new implementation-related routines and hence change their own behaviour deliberately. Providing stronger support for the implementers in the early phases of the intervention might thus be beneficial to enhance implementers’ action-control skills needed for intervention maintenance [82]. In following what some of our implementers intuitively did and what research around implementation intentions and habit formation suggest [81,82,83,84,85,86], implementers could be guided to make detailed plans on integrating implementation tasks into existing routines at the workplace, and to create contextual cues—or choice architectures—that automatically guide them to perform these tasks. Additionally, to further promote habit formation, implementers could be encouraged to perform implementation tasks consistently and regularly [82,84].

Besides providing guidance for forming the implementation into a routine, our data speak for the necessity of a more comprehensive implementer training. The training should ensure everyone involved—including individuals that join the process later—understands the rationale, purpose, and significance of the intervention, how the intervention is assumed to work, and the tasks each implementer is expected to complete. As for the significance, the training should help implementers see the relevance of the intervention for themselves, their work community, and the organisation. Evidence suggests that increased understanding can strengthen motivation [82] and result in improved implementation [41]. Otherwise, implementers may find the intervention personally insignificant, as occurred at some of our intervention sites. Regarding the logic behind expected effects, training implementers—or choice architects—should emphasise the importance of timely and accurate delivery. Choice architecture interventions play with details, and slightly wrong timing or non-optimal placement may make otherwise effective strategies lose their power to guide peoples’ choices for the better [19]. This entails that choice architects need to learn to observe and enhance the choice environment to achieve and maintain a set-up that is capable of triggering healthier behaviours. In terms of implementation tasks, our data pointed out that the training should encourage implementers not to give up if they fail to observe immediate effects. Effects might remain undetected if the intervention works for certain individuals during certain time periods or in specific contexts [87], or if the effects manifest with some delay, as typically happens with priming [88,89]. Overall, the above remarks on knowledge-building reflect the NPT construct coherence, which involves building a shared understanding of the aims, value, importance, and benefits of a new practice, as well as the tasks and responsibilities of everyone involved [76,77]. Similarly, prior implementation research stresses the importance of implementer capacity [39,41], and notes that besides information, implementer training should involve practical on-site coaching [79].

In terms of the implementer, our results suggest that implementation benefits from committed, motivated, inspirational, and organised implementers with job substance, duties, and schedules to which the implementation fits. Similarly, DIT acknowledges the role of influential implementers, or opinion leaders, that resemble other members of the community and act as social models [78]. Such “champions” have the respect of the personnel and can help orchestrate interventions from their adoption to maintenance [41]. The characteristic of being organised relates to the above-discussed skills to reinforce habit formation [82]. Compatibility with work, in turn, replicates results of earlier studies [39].

Our findings indicate that informing personnel of the intervention could facilitate implementation through enhanced employee acceptance. This finding aligns with the results of an interview study on consumer acceptance of nudging, which concluded that increasing consumer awareness and comprehension of nudged decision-making contexts predicts higher acceptability [90]. Fortunately, emerging evidence suggests such informing might not compromise intervention effectiveness [91]. Linking back to the above remarks on the importance of shared decision making and collaboration among all involved parties, this finding on openness raises the question, who do we think the choice architects are, and who should they be? In this work, the researchers and the coordinators and implementers of intervention sites acted as choice architects. Future studies could nevertheless consider broadening this perspective. Besides informing employees of implemented strategies, studies could involve employees in designing these strategies. Such an inclusive approach could enhance the ownership, commitment to, and acceptance of interventions on all levels of organisations; thus facilitating improved and sustained implementation. The shared ownership and understanding of implemented strategies could also enable a shared responsibility of maintaining the commonly constructed choice architecture, further supporting fidelity and maintenance.

### 4.2. Strenghts and Limitations

The strengths of this work include the way that the study bridges theory, scientific evidence, and empirical experiences from stakeholders in the field to a practical, adaptable, and workplace-centred intervention approach for real-world circumstances. In collaboration with participating worksites, intervention content and implementation were contextualised and integrated into the activities of each site, aiming to cause minimal disruption to site operations. This co-creative and contextualised approach was expected to improve implementation quality and reflect better long-term maintenance, as literature [20,39,41,49], the normalisation process theory [76,77], the diffusion of innovations theory [78], and the interactive systems framework [79] suggest. Further strengths include the large and heterogeneous study sample, as well as the systematic, mixed-methods analysis of implementation. This analysis enables us to examine the association between implementation and intervention effectiveness [43], variables that prior research has found to be positively correlated [39,41].

The study has its limitations as well. First, the majority of implemented practical strategies were launched by few intervention sites only. The feasibility evaluation of these strategies is thus limited to a small number of cases, reducing the representativeness of observed findings. Second, although our fidelity evaluation excluded cases with too little data for reliable assessment, some ratings nevertheless build on relatively limited data on intervention delivery. Such less comprehensive data pertain particularly to sites to which we had no direct contact. Consequently, the results warrant cautious interpretation. Third, our implementation and feasibility evaluation were limited to select indicators: applicability to diverse worksites, ease of implementation, required purchases, dose delivered, quality of implementation, and maintenance. Intervention evaluation frameworks, however, feature other elements as well, including intervention adoption [42,92]; design, protocol, and implementer training [44,45,47]; intervention reach [42,92], as well as receipt and participant enactment [44,47]. We omitted the evaluation of intervention design, protocol, adoption (i.e., proportion of sites adopting the intervention), and implementer training due to limited resources and space. Yet, we have reported on and discussed these domains in the manuscript. Intervention receipt, which reflects the extent to which study subjects demonstrate knowledge and skills acquired in the intervention [47], was excluded from the analysis because choice architectural interventions do not rely on education and knowledge acquisition [18,30]. Reach refers to the proportion of the target audience that is aware of the intervention [39], and participant enactment implies whether study subjects apply skills learned in the intervention in their daily lives [47]. We consider these dimensions to reflect intervention effects, which we will report elsewhere.

### 4.3. Implications for Practice and Research

The choice architecture of living environments substantially influences dietary behaviour and physical activity. Efforts are hence needed to develop choice architectures that are conducive to healthier behaviours. Workplaces provide one suitable setting for such efforts. The hands-on instrument developed in this study, the StopDia Toolkit for Creating Healthy Working Environments, portrays a broad selection of practical, evidence-based, fairly effortless, and inexpensive choice architectural strategies for several generic settings in the workplace. For effective implementation, we recommend adapting the strategies to local contexts and considering the facilitators and barriers detailed in this paper. To build necessary capacity for implementation, organisations typically need support from external partners [41,79], such as the research team in the current study. In future, occupational wellbeing and health service providers or other organisations working for occupational and public health could be apt partners for providing the support. Moreover, although this study focused on workplaces, its contribution could benefit other real-world settings as well, such as schools, grocery shops, and catering services. Future research is needed to confirm our findings and to increase understanding of, inter alia, the following topics: (1) the effects, (2) the association between implementation and effects, (3) the acceptance, (4) the full implementation costs, and (5) the relationship between costs and effects of choice architecture interventions implemented in real-world settings.

## 5. Conclusions

Our findings suggest that a broad range of choice architectural strategies for healthier dietary choices and physical activity are applicable to diverse workplaces. These strategies fit generic workplace choice architectures, but tailoring to local contexts, i.e., contextualisation, improves their feasibility and implementation. Collaboration with intervention sites is thus recommended when designing real-world implementation; considering the characteristics of the organisation, intervention, worksite environment, and implementer. Sufficient training and support for implementers, as well as management support appear important for sustained and high-quality implementation.

## Figures and Tables

**Figure 1 nutrients-13-03592-f001:**
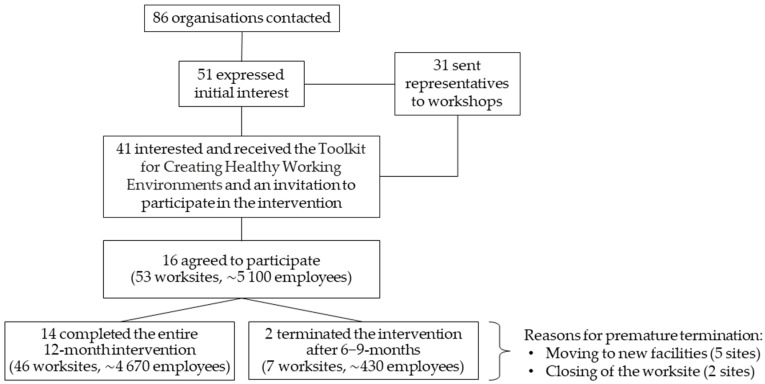
Flow chart of the recruitment and participation of organisations. Numbers refer to organisations, unless otherwise specified.

**Figure 2 nutrients-13-03592-f002:**
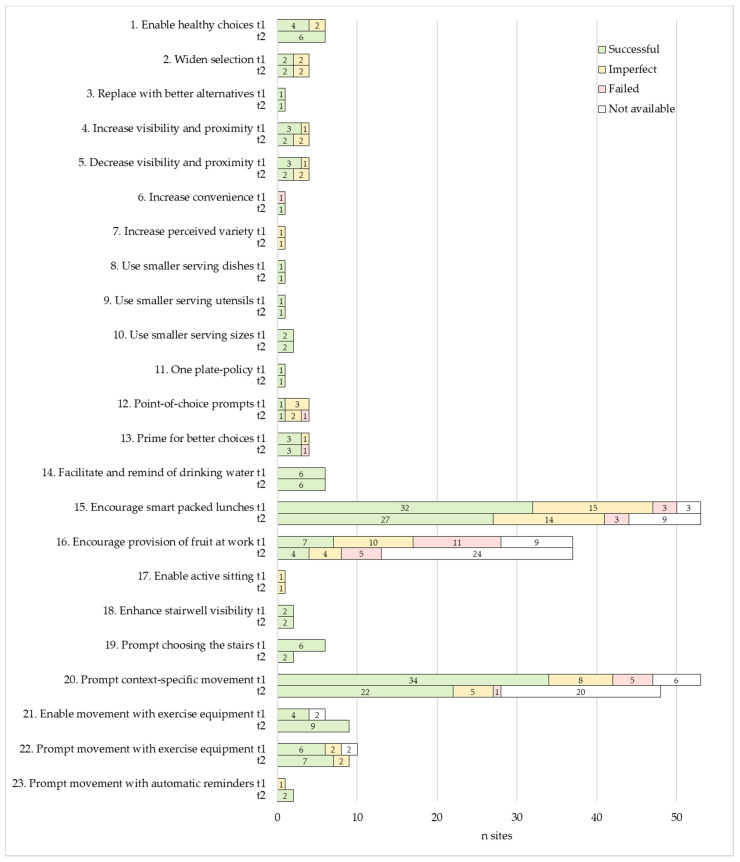
Implementation quality ratings by practical strategy at month 6 (t1) and month 12 (t2).

**Figure 3 nutrients-13-03592-f003:**
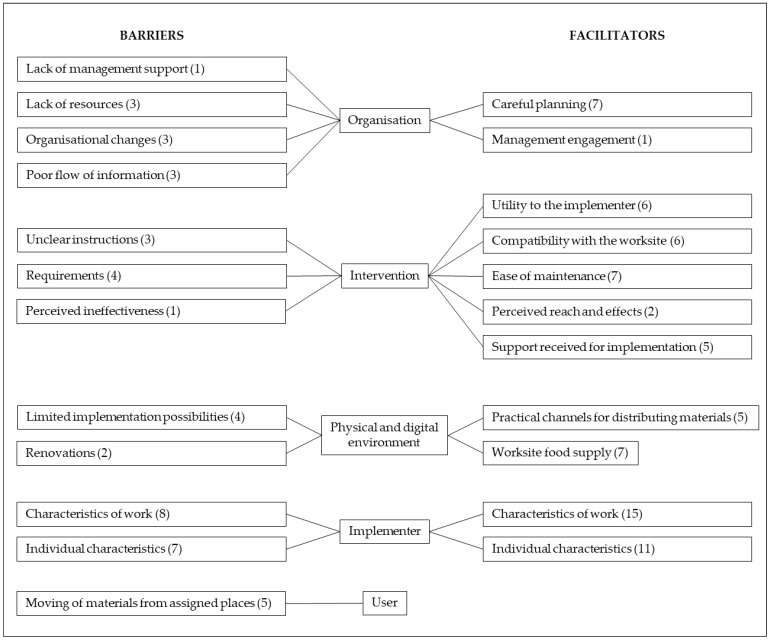
Main categories of the facilitators and barriers of implementation identified through qualitative content analysis. Numbers refer to organisations associated with each category.

**Table 1 nutrients-13-03592-t001:** Characteristics of participating organisations.

Organisation	Sector	Field of Operation	*n* Sites	*n* Employees ^1^	% Men	Type of Work	Shift Work
O1	Private	Retail	5	360	21	Mixed ^2^	Yes
O2	Private	Metal industry	1	600	80	Mixed ^2^	Yes
O3	Private	Forest industry	1	950	78	Mixed ^2^	Yes
O4	Private	Retail	3	300	20	Mixed ^2^	Yes
O5	Private	Higher education	5	370	34	Sedentary	No
O6	Public	Municipality	1	70	29	Sedentary	No
O7	Private	Chemical industry	1	400	75	Mixed ^2^	Yes
O8	Private	Farming	1	140	35	Mixed ^2^	Yes
O9	Public	Municipality	1	80	39	Sedentary	No
O10	Public	Municipality	3	250	32	Mixed ^2^	Yes
O11	Private	Construction industry	5	180	91	Mixed ^2^	No
O12	Public	Health care	20	490	46	Mixed ^2^	Yes
O13	Private	Food industry	1	250	70	Mixed ^2^	Yes
O14	Private	Retail	3	320	18	Mixed ^2^	Yes
O15	Public	Municipality	1	300	20	Sedentary	No
O16	Public	Welfare services	1	40	5	Mixed ^2^	No

^1^ Approximate number of employees exposed to the intervention, ^2^ a mixture of physical and sedentary work.

**Table 2 nutrients-13-03592-t002:** Characteristics of strategies implemented and the number (*n*) of intervention sites that implemented each strategy. The total number of sites was 53.

Target	Practical Strategy	Description	Behaviour Change Mechanism ^3^	Ease of Implementation ^4^	Required Purchases ^4^	Setting	*n*
	**Food provision**						
Nutrition	Enable healthy choices	Healthy ^1^ food and beverage choices, such as fruit and smoothies made available.	Product availability ↑ ^T^	Moderate	Minor	Meetings	6
Nutrition	2. Widen selection	Greater variety of healthy ^1^ food and beverage options available.	Product availability ↑ ^T^ Attractive (salience ↑) ^M, E^	Moderate	Minor	Cafeteria	4
Nutrition	3. Replace with better alternatives	Energy dense and nutritionally poor options replaced with similar but nutritionally better alternatives.	Product availability ↑ ^T^ Easy (substitution, default) ^E^	Moderate	Minor	Meetings	1
Nutrition	4. Increase visibility and proximity	Healthy ^1^ options placed: (a) in visible spots, (b) at the beginning of the buffet, (c) closer to the chooser (e.g., in front row), and/or (d) in the middle of the tray, shelf, or showcase.	Product position ^T^ Easy (friction costs ↓) ^E^ Attractive (salience ↑) ^M, E^	Demanding	None	Cafeteria	4
Nutrition	5. Decrease visibility and proximity	Less healthy options placed: (a) in less visible spots, (b) at the end of the buffet, (c) further away from the chooser (e.g., in back row), and/or (d) on the edge of the tray, shelf, or showcase.	Product position ^T^ Less easy (friction costs ↑) ^E^ Less attractive (salience ↓) ^M, E^	Demanding	None	Cafeteria	4
Nutrition	6. Increase convenience	Fruit and vegetable served ready to eat, i.e., washed, peeled if needed, and cut into pieces.	Product functionality ^T^ Easy (friction costs ↓) ^E^	Demanding	None	Meetings	1
Nutrition	7. Increase perceived variety	Salad components served from separate serving dishes to encourage greater consumption.	Product presentation ^T^ Attractive (salience ↑) ^M, E^	Moderate	None	Cafeteria	1
Nutrition	8. Use smaller serving dishes	Less healthy foods served from smaller serving dishes.	Product size ^T^ Easy (default) ^M, E^	Moderate	None	Cafeteria	1
Nutrition	9. Use smaller serving utensils	Less healthy foods served with smaller tongs and spoons.	Product size ^T^ Less easy (default) ^M, E^	Moderate	None	Cafeteria	1
Nutrition	10. Use smaller serving sizes	Less healthy options served in smaller sizes.	Product size ^T^ Easy (default) ^M, E^	Moderate	None	Meetings	2
Nutrition	11. One plate-policy	Separate bread and salad plates moved out of sight to guide employees to choose one large plate; thus facilitating the composition of the meal according to the plate model (i.e., 1/2 vegetable, 1/4 protein, and 1/4 carbohydrates). For the strategy to be effective, salads should be placed first in the buffet line.	Product size ^T^ Easy (default) ^M, E^	Easy	None	Cafeteria	1
Nutrition	12. Point-of-choice prompts	Healthy ^1^ options indicated with the Heart Symbol ^2^ on menus and at the point of choice.	Information on related objects ^T^ Attractive (salience ↑) ^M, E^ Timely (prompting) ^E^ Easy (simplification) ^E^	Demanding	None ^5^	Cafeteria	4
Nutrition	13. Prime for better choices	Follow the heart posters ^2^ at restaurant entrance and/or at the beginning of the buffet to guide customers to notice and choose options labelled with the Heart Symbol ^2^.	Information within the wider environment ^T^ Attractive (salience ↑) ^M, E^ Timely (priming) ^M, E^	Easy	None ^5^	Cafeteria	4
	**Drinking water**						
Nutrition	14. Facilitate and remind of drinking water	Personal, reusable water bottles provided for employees.	Related object availability ↑ ^T^ Easy (friction costs ↓) ^E^	Easy	Minor	Personal workstation	6
	**Packed lunches and snacks**						
Nutrition	15. Encourage smart packed lunches	Temptingly named, visually attractive, and seasonal StopDia packed lunch of the week recipes ^2^ promoted and provided at workplace coffee rooms and/or via electronic channels, such as info-screens, company intranet, and newsletters. The campaign comprises one recipe for each week of the year, and all recipes meet the nutritional criteria of the Heart Symbol ^1^.	Easy (friction costs ↓, chunking) ^E^ Attractive (salience ↑) ^M, E^ Social (descriptive norm) ^M, E^ Timely (priming) ^M, E^ Affect ^M^	Moderate	None ^5^	Coffee rooms	48
Nutrition	16. Encourage provision of fruit at work	The promotion and provision of the fruit crew starting kit ^2^ that facilitates colleagues to found a fruit circle and consequently have fresh fruit available at the workplace.	Social (network nudge, commitment contracts, descriptive norm, reciprocity) ^M, E^ Attractive (gamification, salience ↑) ^M, E^ Timely (implementation intentions) ^E^	Easy	None ^5^	Coffee rooms	17
	**Time spent sitting**						
Physical activity	17. Enable active sitting	Introduction of alternative seats, such as wobble chairs or balance cushions.	Product availability ^T^ Easy (friction costs ↓) ^E^	Easy	Substantial	Common environments	1
	**Stair use**						
Physical activity	18. Enhance stairwell visibility	Footprints attached on the floor to lead to stairs from the point of choice between the stairs and the elevator.	Atmospheric properties of the wider environment ^T^ Attractive (salience ↑) ^M, E^ Timely (prompting) ^E^	Easy	None ^5^	Elevator, stairs	2
Physical activity	19. Prompt choosing the stairs	StopDia logo (a stop hand sign with a heart on the palm) ^2^ placed on elevator doors, next to elevator call buttons, or in their immediacy.	Timely (prompting) ^E^	Easy	None ^5^	Elevator	6
	**Movement breaks**						
Physical activity	20. Prompt context-specific movement	StopDia Flex! movement posters ^2^ placed on salient spots where employees typically pause for a moment and have the opportunity to perform movements. Such spots can be, for example, by copy machines, microwaves, kettles, coffee makers, and bathrooms.	Timely (prompting) ^E^ Attractive (salience ↑) ^M, E^ Easy (chunking) ^E^	Easy	None ^5^	Common environments	43
Physical activity	21. Enable movement with exercise equipment	Light exercise equipment made available, for example, gym sticks, balance boards, or hanging bars.	Product availability ^T^ Easy (friction costs ↓) ^E^	Easy	Minor	Common environments	9
Physical activity	22. Prompt movement with exercise equipment	Available exercise equipment placed on salient spots where employees typically pause for a moment, and an opportunity for a short exercise break occurs, for example, by copy machines, micros, kettles, or coffee makers.	Timely (prompting) ^E^ Attractive (salience ↑) ^M, E^	Moderate	None	Common environments	13
Physical activity	23. Prompt movement with automatic reminders	An application that prompts to take short exercise breaks at pre-set intervals provided for employees.	Timely (prompting) ^E^	Easy	Minor	Personal workstation	2

^1^ Food products, meals, and recipes that meet the product category-specific nutritional criteria of the Heart Symbol (https://www.sydanmerkki.fi/en/ (accessed on 12 October 2021)), as well as energy-free beverages; ^2^ for images of the materials, see Appendix A; ^3^ behaviour change mechanisms: ^T^ = TIPPME [56], ^M^ = MINDSPACE [57], ^E^ = EAST [58], ↑ = increase, ↓ = decrease; ^4^ for definitions of categories, see Appendix A; ^5^ the study treated and delivered needed materials.

**Table 3 nutrients-13-03592-t003:** The associations of implementation quality (0 = failed, 1 = imperfect, 2 = successful) at month 6 (t1) and month 12 (t2), with the ease of implementation of applied strategies and the three diverse modes of support that the research team could provide.

Independent Variable	t1				t2			
	*n* Cases	Mean	95% CI for Mean	*p* ^1^	*n* Cases	Mean	95% CI for Mean	*p* ^1^
Ease of implementation								
Easy	100	1.47	1.32–1.62	0.535 ^2^	68	1.65	1.49–1.81	0.187 ^2^
Moderate	74	1.62	1.49–1.75		69	1.64	1.50–1.77	
Demanding	13	1.46	1.06–1.86		13	1.38	0.99–1.78	
Researcher assisted intervention launch								
Yes	63	1.71	1.59–1.84	0.021 ^3^	54	1.59	1.42–1.76	0.625 ^3^
No	124	1.44	1.30–1.57		96	1.64	1.51–1.76	
Direct contact to intervention site								
Yes	127	1.68	1.59–1.77	0.000 ^3^	117	1.64	1.54–1.74	0.980 ^3^
No	60	1.22	0.99–1.44		33	1.55	1.26–1.83	
SMS reminders for strategy 15								
Yes	12	1.83	1.59–2.08	0.100 ^3^	12	1.75	1.46–2.04	0.290 ^3^
No	38	1.50	1.29–1.71		32	1.47	1.23–1.71	

^1^ *p*-values < 0.05 statistically significant; ^2^ Kruskall–Wallis test; ^3^ Mann–Whitney U test.

## Data Availability

The data presented in this study, including quantitative data and qualitative Finnish language data, will be made available on request from the corresponding author. The data are not publicly available due to privacy concerns.

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
