# Peer review of "Choice Architecture Cueing to Healthier Dietary Choices and Physical Activity at the Workplace: Implementation and Feasibility Evaluation"

_nutrients, 2021, doi:10.3390/nu13103592_

Round 1
Reviewer 1 Report
I suggest to present not only the results of StopDia process adherence but the diabetes prevention results of the participants in StopDia study (Ex. the number of study dropout, the level of fasting glucose and HbA1C, etc.).Author Response
Thank you for the suggestion to include in the paper individual-level effectiveness results of the intervention.
Since the StopDia at Work intervention targeted the worksite environment, we recruited worksites instead of individual employees. Hence, in the current study, the worksites represent study participants.
Regarding dropouts, no worksite dropped out the study during the 12-month intervention. A few worksites terminated the intervention early, i.e. after 6-9 months, due to the moving of the worksites into new facilities or due to the closing of the worksites. We have reported the number of these sites and the reasons for premature termination in Figure 1 and in the Materials and methods section 2.5.1.
We did not include effectiveness evaluation in the paper currently under review, since the aim of the paper is to evaluate the implementation and feasibility of the StopDia at Work intervention, focusing on the fidelity, facilitators and barriers, and maintenance of implementation. However, we aim to publish a separate paper on the effects of the intervention. This effectiveness paper will build on questionnaire data collected from the employees of intervention worksites. Unfortunately, we did not collect blood samples from these employees. Hence, we cannot report fasting glucose or HbA1C.
Reviewer 2 Report
Thank you so much for inviting me to review this manuscript. Authors tried to promote healthy dietary choices and daily physical activity at the workplace with subtle modifications to the worksite environment, including common working spaces, personal workstations, recreation rooms, stairwells, elevators, and cafeterias. Congratulations to the authors for this laborious work.
Minor comments:
My main concern is about the amount of references used (especially in the introduction). In my opinion, there are too many. There are aspects that can be explained with one or two references.
On the other hand, abstract should include the aim of the study.
Best wishes,
Author Response
Thank you for the positive comment. We reviewed all references and reduced the ones not essential. Please find below the list of references removed.
In addition, we rephrased the second sentence of the abstract as follows: "The aim of this paper is to portray the implementation and feasibility assessment of a 12-month mixed-methods intervention study, StopDia at Work, targeting the environment of 53 diverse worksites."
References removed from the introduction
Glanz, K.; Sallis, J.F.; Saelens, B.E.; Frank, L.D. Healthy Nutrition Environments: Concepts and Measures. American Journal of Health Promotion 2005, 19, 330–333, doi:10.4278/0890-1171-19.5.330.
Story, M.; Kaphingst, K.M.; Robinson-O’Brien, R.; Glanz, K. Creating Healthy Food and Eating Environments: Policy and Environmental Approaches. Annual Review of Public Health 2008, 29, 253–272, doi:10.1146/annurev.publhealth.29.020907.090926.
Swinburn, B.; Egger, G.; Raza, F. Dissecting Obesogenic Environments: The Development and Application of a Framework for Identifying and Prioritizing Environmental Interventions for Obesity. Prev Med 1999, 29, 563–570.
Papies, E.K. Health Goal Priming as a Situated Intervention Tool: How to Benefit from Nonconscious Motivational Routes to Health Behaviour. Health Psychology Review 2016, 10, 408–424, doi:10.1080/17437199.2016.1183506.
Marteau, T.M.; Hollands, G.J.; Fletcher, P.C. Changing Human Behavior to Prevent Disease: The Importance of Targeting Automatic Processes. Science 2012, 337, 1492–1495, doi:10.1126/science.1226918.
Sheeran, P. Intention—Behavior Relations: A Conceptual and Empirical Review. European Review of Social Psychology 2002, 12, 1–36, doi:10.1080/14792772143000003.
Lorenc, T.; Petticrew, M.; Welch, V.; Tugwell, P. What Types of Interventions Generate Inequalities? Evidence from Systematic Reviews. Journal of Epidemiology and Community Health 2013, 67, 190–193, doi:10.1136/jech-2012-201257.
He, C.; Breiting, S.; Perez-Cueto, F.J.A. Effect of Organic School Meals to Promote Healthy Diet in 11-13 Year Old Children. A Mixed Methods Study in Four Danish Public Schools. Appetite 2012, 59, 866–876, doi:10.1016/j.appet.2012.09.001.
Zlatevska, N.; Dubelaar, C.; Holden, S.S. Sizing up the Effect of Portion Size on Consumption: A Meta-Analytic Review. Journal of Marketing 2014, 78, 140–154, doi:10.1509/jm.12.0303.
Bellicha, A.; Kieusseian, A.; Fontvieille, A.-M.; Tataranni, A.; Charreire, H.; Oppert, J.-M. Stair-Use Interventions in Worksites and Public Settings - A Systematic Review of Effectiveness and External Validity. Preventive Medicine 2015, 70, 3–13, doi:10.1016/j.ypmed.2014.11.001.
Fernandes, A.C.; Oliveira, R.C.; Proenca, R.P.C.; Curioni, C.C.; Rodrigues, V.M.; Fiates, G.M.R. Influence of Menu Labeling on Food Choices in Real-Life Settings: A Systematic Review. Nutrition Reviews 2016, 74, 534–548, doi:10.1093/nutrit/nuw013.
Forberger, S.; Reisch, L.; Kampfmann, T.; Zeeb, H. Nudging to Move: A Scoping Review of the Use of Choice Architecture Interventions to Promote Physical Activity in the General Population. International Journal of Behavioral Nutrition and Physical Activity 2019, 16, 77, doi:10.1186/s12966-019-0844-z.
Glasgow, R.E.; Vogt, T.M.; Boles, S.M. Evaluating the Public Health Impact of Health Promotion Interventions: The RE-AIM Framework. American Journal of Public Health 1999, 89, 1322–1327.
Pronk, N.P. Designing and Evaluating Health Promotion Programs. Disease Management & Health Outcomes 2003, 11, 149–157, doi:10.2165/00115677-200311030-00002.
References changed in Discussion
We replaced the following three references:
Kroese, F.M.; Marchiori, D.R.; de Ridder, D.T.D. Nudging Healthy Food Choices: A Field Experiment at the Train Station. Journal of Public Health 2016, 38, e133–e137, doi:10.1093/pubmed/fdv096.
Bruns, H.; Kantorowicz-Reznichenko, E.; Klement, K.; Luistro Jonsson, M.; Rahali, B. Can Nudges Be Transparent and yet Effective? Journal of Economic Psychology 2018, 65, 41–59, doi:10.1016/j.joep.2018.02.002.
Cheung, T.T.L.; Gillebaart, M.; Kroese, F.M.; Marchiori, D.; Fennis, B.M.; de Ridder, D.T.D. Cueing Healthier Alternatives for Take-Away: A Field Experiment on the Effects of (Disclosing) Three Nudges on Food Choices. BMC Public Health 2019, 19, 974, doi:10.1186/s12889-019-7323-y.
… with the following review:
de Ridder, D.; Kroese, F.; van Gestel, L. Nudgeability: Mapping Conditions of Susceptibility to Nudge Influence. Perspectives on Psychological Science 2021, 174569162199518, doi:10.1177/1745691621995183.